# A Study on the Influence of the Electroplating Process on the Corrosion Resistance of Zinc-Based Alloy Coatings

Fan Cao [1], Jianpeng Wang [2], Yuli Lian [3], Yuanhao Wang [1,4], Xue Wang [1], Xiaomin Wang [5], Aiqing Song [6] and Lei Shi [1,*]

[1] School of Materials Science and Engineering, Shandong Jianzhu University, Jinan 250101, China; caofan_success@126.com (F.C.)

[2] School of Materials and Chemical Engineering, Southwest Forestry University, Kunming 650224, China; ddfdarkholme@126.com

[3] Department of Biochemical Engineering, Chaoyang Teachers College, Chaoyang 122000, China

[4] Yangzhou Hanshun Automotive Components Co., Ltd., Yangzhou 225000, China

[5] Jinan Institute of Quantum Technology, Jinan 250000, China

[6] Wendeng Guangrun Metal Products Co., Ltd., Weihai 266440, China

[*] Correspondence: slcqj@sdjzu.edu.cn

**Abstract:** A comprehensive analysis was conducted to examine the crystal phase composition, surface and cross-section morphology, elemental composition, thickness, and corrosion resistance of coatings. X-ray diffraction (XRD) was employed to investigate the texture and crystal phase of the materials while scanning electron microscopy (SEM) and energy-dispersive spectroscopy (EDS) were utilized to assess the surface and cross-section structure. Additionally, electrochemical techniques were employed to evaluate the corrosion performance. Compared to DC electroplating, the corrosion potential of pulsed galvanized ferroalloy alloy coating increased from $-1031$ mV to $-1008$ mV, and the corrosion current density decreased from $3.122 \times 10^{-5}$ A·cm$^{-2}$ to $0.321 \times 10^{-5}$ A·cm$^{-2}$. Moreover, the corrosion rate value of the coating obtained by the pulse rectifier ($0.386 \times 10^{-5}$ g m$^{-2}$ h$^{-1}$) was lower than that obtained by the DC power supply ($3.75 \times 10^{-5}$ g m$^{-2}$ h$^{-1}$). Additionally, pulsed electrodeposition reduced the iron content of the coating by 0.7%, thereby enhancing its corrosion resistance and flatness. The impedance parameters of the zinc–iron alloy coating acquired through the 30% duty cycle monopulser process exhibit superior performance compared to DC electroplating. Evidently, the monopulse coating's structure enhances crystal packing density, augments coating thickness, improves adhesion to the substrate interface, and optimizes grain distribution uniformity. These factors are crucial determinants of the corrosion behavior exhibited by Ze–Fe coating.

**Keywords:** Zn–Fe alloy; monopulse electroplating; corrosion resistance

## 1. Introduction

Electrodeposition is a preferred coating technique due to scalability and low cost [1]. Electroplating is a chemical process of depositing certain metal ions on the surface of alloys and other materials (such as plastics) by electrolysis to improve the ability of wear resistance [2,3], conductivity [4,5], reflectivity [6,7], anti-corrosion [8,9], etc. Conventional traditional electroplating is generally DC electroplating, and in recent years, with the improvement of the corrosion resistance and decorative requirements of enterprises, as well as the rapid development of pulse electrodeposition (PED), the traditional electroplating method has been unable to meet social needs. Therefore, the application range of pulse plating is getting wider and wider, which promotes the improvement of corrosion resistance and the performance of coatings.

Zinc and its composites/alloys can be electrodeposited by the pulse current method [10–12]. However, the electrodeposition of zinc and zinc alloys has been widely used in many industries, such as door and window hardware, automotive, and aircraft. Zn coating prevents the corrosion

of steel substrate in two ways [13,14]: the barrier and the galvanic protection have unique advantages over most other finishing technologies [15]. The Zn-1 wt.% Fe and Zn-10 wt.% Fe alloy sub-layers consisted of two phases: the $\eta$-phase (Zn–Fe solid solution) and the $\delta$1-phase (intermetallic $FeZn_7$) [16]. It could be suggested that the electrodeposited Zn–Fe multilayered alloy coatings do not follow the Zn–Fe equilibrium phase diagram, confirming previous results that support the predominance of non-equilibrium conditions during the electrodeposition of Zn–Fe alloy coatings [17–20]. The Zn–Fe alloy is a frequently utilized zinc alloy [21–23], which has the characteristics of high corrosion resistance and low cost. Growth conditions have a considerable impact on the surface properties of coatings [24]. Warfsmann et al. [25] reported on stoichiometrically varied Zn–Fe films prepared by electrochemical deposition, which can potentially be used as photoanodes in solar water-splitting cells.

Arrighi et al. [26] studied the effect of high iron content in pulse electrogalvanized ferroalloy on coatings in an additive-free gluconate-based electrolyte. The results show that the coating produced by pulse electroplating reduced the impact of the hydrogen evolution reaction (HER). Notably, the composition and microstructure of the alloys were significantly affected by the frequency and current density. Specifically, an increase in pulse frequency from 50 Hz to 1000 Hz resulted in a decrease in Zn content from 7.2 wt.% to 2.1 wt.% and coarsening of grains [27]. The influence of pulse plating parameters on the chemical and phase composition, surface topography, and corrosion resistance of Zn–Co (<1%) and Zn–Fe (<1%) alloy coatings have been studied [28]. It was suggested that the grain size of pulsed galvanizing Co and zinc–iron coatings for zinc alloys was reduced from 61 nm to 36 nm and from 72 nm to 50 nm, respectively, compared to DC plating. The dual-phase formation was the principal reason pulse-plated low-alloyed Zn coatings did not exhibit any improvement in their corrosion resistance [29].

Shourgeshty et al. [30] utilized the pulsed electrodeposition technique to deposit two distinct functional gradient Zn–Ni–$Al_2O_3$ coatings. The first coating exhibited an increase in Ni, alumina content, and microhardness towards the surface. Conversely, the second coating demonstrated an increase in corrosion rate within the range of 2.9 to 8.7 $\mu A \cdot cm^{-2}$ while also exhibiting improved wear resistance. Ni–TiC nanocomposites were synthesized on a Q235 steel matrix using magnetic-assisted pulsed electrodeposition (MPED) technology. The coatings obtained through pulsed electrodeposition exhibit small Ni and TiC grains, high TiC content, and microhardness [31]. Additionally, Ni–Co/$SiO_2$ nanomaterials can be produced via Watt's bath using both pulsed and DC deposition techniques. It has been demonstrated that the microhardness of pulsed electrodeposition nanomaterials surpasses that of direct electrodeposition nanomaterials [32]. The pulsed electrodeposition method can be utilized to synthesize a three-dimensional Ni–Fe–P electrocatalyst on nickel nanostructures, resulting in nanostructures that demonstrate exceptional intrinsic electrocatalytic activity. Moreover, the electrodes prepared from these nanostructures exhibit favorable electrocatalytic activity and stability [33]. Additionally, the pulsed current electrodeposition technique can be employed to fabricate Ni–W–SiC alloy nanofilms with fine crystal structures, smooth and uniform surfaces, and excellent corrosion resistance [34].

Krajaisri et al. [35] conducted a study to investigate the impact of the pulse duty cycle and electrolyte temperature on the electric deposition of a Sn–Cu alloy. They compared this method with DC electroplating and demonstrated that the coating obtained through pulse electroplating exhibited superior corrosion resistance and flatness. In a similar vein, Vamsi et al. [36] prepared crack-free nanocrystalline Ni–W alloy coatings using pulsed current electrodeposition. They extensively characterized the deposited and heat-treated coatings and provided evidence that the pulsed process can enhance the mechanical properties of the coatings.

The primary objective of this study is to modify the formulation and monopulse process of the electrodeposited Zn–Fe alloy plating in order to enhance and optimize the flatness, corrosion resistance, hardness, and binding strength of the coating. Detailed discussions and comparisons are conducted on the crystal phase, surface and cross-sectional

morphology, elemental composition, and thickness of coatings obtained through monopulse and DC processes. Additionally, an attempt is made to evaluate the relationship between process parameters, structural characteristics, and corrosion resistance.

## 2. Materials and Methods

### 2.1. Materials

Based on prior patent advancements (as shown in the patents section), the bath's efficacy was enhanced. $ZnCl_2$ serves as the primary salt in the bath due to the chloride ion's small ionic radius, potent penetration capabilities, and substantial adsorption force on metal surfaces. However, an abundance of chloride ions can produce soluble chloride on the coating, leading to an increased occurrence of pitting defects. Furthermore, in the plating process, the customary monodentate ligand chloride ion expedites the metal deposition rate on the cathode, leading to an augmentation of metal particles and an uneven coating surface. Consequently, this research employed ZnO as the primary salt in the bath, instead of $ZnCl_2$, to mitigate the adverse impact of chloride ions to a greater degree.

All chemicals used in the experiment are analytically pure including NaOH (analytically pure, Tianjin, China), ZnO (analytically pure, Tianjin, China), $FeCl_2$ (analytically pure, Tianjin, China), ascorbic acid (analytically pure, Sinopharm Group), EDTA-$Na_2$ (analytically pure, Tianjin, China), DPE-III (analytical purity, Jinan, China), BH-336 (analytical purity, Jinan, China), $HNO_3$ (analytical purity, Qingdao, China), NaCl (analytical purity, Qingdao, China), HCl (analytical purity, Qingdao, China), and potassium sodium tartrate (analytical purity, Tianjin, China).

DPE-III serves as the principal brightener in alkaline zincate galvanizing, comprising primarily of dimethylaminopropylamine. BH-336, on the other hand, functions as the secondary brightener, encompassing carrier brightener, main brightener, auxiliary brightener, wetting agent, and impurity removal agent. These two additives are simultaneously introduced into the plating solution, working in conjunction with other components to refine the coating grains, resulting in a more uniform, denser, and flatter coating microstructure.

The performance of the coating is compared with the DC and the pulse rectifier; the DC one is a KSY-multi-function experimental rectifier, and the pulse one is an SMD-type intelligent multi-pulse electroplating rectifier. The 45# mild steel with dimensions of 45 mm × 40 mm × 2 mm was made into the anode and the cathode. The mild steel sheet is sanded and polished with 400#, 800#, and 1000# sandpapers before electroplating and then washed thoroughly with absolute ethanol, acetone, and distilled water for 3 min.

### 2.2. Formulation and Methodology

All chemicals used to prepare electroplating samples are analytical-grade chemicals acquired from different manufacturers in China. Electrolytes were prepared using ZnO (10–14 g/L), NaOH (100–150 g/L), $FeCl_2$ (1–2 g/L), ascorbic acid (15–20 g/L), EDTA-$Na_2$ (15–20 g/L), DPE-III (5–10 mL/L), and BH-336 (2–6 mL/L). The weighed NaOH and ZnO are dissolved completely by stirring in distilled water and placed in the same beaker. $FeCl_2$, ascorbic acid, and EDTA-$Na_2$ were dissolved at the same time to prevent $Fe^{2+}$ oxidation into $Fe^{3+}$ at the same time, and $Fe^{2+}$ was fully complex. Finally, DPE-III and BH-336 were poured into the reagent sequentially. Annealed mild steel (45 mm × 40 mm × 2 mm) was used as the cathode and anode, respectively. The electrodes before electrode deposition were polished and cleaned before plating to ensure the combination of the coating and the substrate. Under the condition that the current density was 9 mA/cm$^2$ and the room temperature was 25 °C, the DC and multi-pulse rectifier were used for electrodeposition for 15 min, respectively. The compositions and parameters selection of monopulse and DC electrodeposition are listed in Tables 1 and 2, respectively.

**Table 1.** Composition and parameter selection for monopulse electrodeposition of Zn–Fe alloys.

| Composition and Parameter Selection | Concentration and Parameter Value |
|---|---|
| ZnO (g/L) | 10–14 |
| NaOH (g/L) | 100–150 |
| FeCl$_2$ (g/L) | 1–2 |
| Ascorbic acid (g/L) | 15–20 |
| EDTA-Na$_2$ (g/L) | 15–20 |
| DPE-III (ml/L) | 5–10 |
| BH-336 (ml/L) | 2–6 |
| Temperature (°C) | 25 |
| pH | 12 |
| Mean current density (A/dm$^2$) | 9 |
| Duty cycle (r) | 30% |
| Work time (min) | 15 |
| Frequency (Hz) | 1000 |

**Table 2.** Parameter selection for DC electrodeposition of Zn–Fe alloys.

| Parameter Selection | Parameter Value |
|---|---|
| Temperature (°C) | 25 |
| pH | 12 |
| Mean current density (A/dm$^2$) | 9 |
| Working time (min) | 15 |

In a monopulse electrodeposition experiment, the pulse frequency of the applied monopulse rectangular, square wave [i.e., $f = 1/(t_{on} + t_{off})$] and the duty cycle [i.e., $\theta = t_{on}/(t_{on} + t_{off})$] remained unchanged. The pulse on time ($t$) varied between 25 and 200 ms, and the peak current ($I_{peak}$) remained between 4 A and 6 A. The application method was used to increase the current density to the average current ($I_{avg}$) for a duration of $t_{on}$, and then the current density was reduced to 0 for a duration of $t_{off}$, and the process was a cycle. Monopulse electrodeposition waveforms and specific processes are shown in Figure 1. The $t_{on}$ is the pulse on time (or pulse width), $t_{off}$ is the pulse off time, $I_{peak}$ represents the peak current, $I_{avg}$ is the average current, and $T$ is the pulse on–off period, that is, $T = t_{on} + t_{off}$. The process parameters are duty cycle $\gamma = 30\%$, frequency $f = 1000$ Hz, average current density $i_{avg} = 9$ A/dm$^2$, and plating time $t = 10$ min.

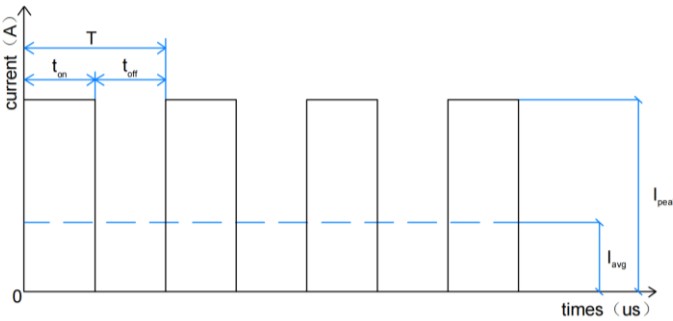

**Figure 1.** Monopulse electrodeposition waveform of the Zn–Fe alloy coating.

At a reduced duty cycle, the nucleation rate of the coating exhibits an increase while the grain size of the sediment experiences a decrease. Nevertheless, the impact of the duty cycle on the coating's properties cannot be accurately foreseen for particular systems due to the occurrence of adsorption/desorption during crystallization, which is significantly influenced by the bath's composition. The production of fine grains during the actual process relies on the conditions prevailing during the interruption of the current. This process facilitates the desorption of impurities and the subsequent formation of fine grains to stimulate renucleation.

Several studies have demonstrated that a coating with a 30% duty cycle exhibits superior surface morphology and corrosion resistance in various systems. For instance, the Zn–Ni–$Al_2O_3$ composite coating displays enhanced delicacy and smoothness at a 33% duty cycle, while the deposition of Ni–Mo alloy on 316 L stainless steel yields the highest activated alloy coating at a 30% duty cycle. Additionally, the nanostructured titanium aluminum nitride coating, prepared through plasma-assisted chemical vapor deposition (PACVD), exhibits the best corrosion resistance at a 30% duty cycle [37–39].

### 2.3. Electrochemical Tests

The Tafel and electrochemical impedance spectroscopy (EIS) of the prepared samples were measured at 25 °C using the Shanghai Chenhua CHI760E electrochemical workstation in accordance with ASTM G5-14 and ASTM G106-89 standards. Among them, the measurement used a standard three-electrode system, incorporating the sample with an exposed area of 1 $cm^2$ as the working electrode, the platinum electrode (Pt) as the auxiliary electrode, and the saturated calomel electrode as the reference electrode. The electrolyte system of the test sample was 3.5% NaCl solution. The open circuit potential was monitored for 10 min. Next, electrochemical impedance spectroscopy (EIS) measurements were performed with an amplitude of 5 mV at a frequency range of 100 kHz to 0.1 Hz. After EIS, the potentiodynamic polarization test (Tafel) was performed at a scanning speed of 1 mV/s. It should be emphasized that the testing sequence of Tafel and EIS should not be reversed because the Tafel test will affect the surface state of the passivation film, resulting in inaccurate EIS test results. The test should be conducted several times to ensure the reproducibility of the experimental results.

### 2.4. Microstructure Characterization

Using a ZEISS Ultra™ Model 55 Scanning Electron Microscope (SEM) with Secondary Electron signal (SE), the surface morphology of the coatings was observed at the acceleration voltage of 5–10 kV. The chemical compositions of the coatings were analyzed using Energy Dispersion Spectroscopy (EDS) based on the SEM, and the EDS results were semi-quantitative. The phase composition of the samples was detected by a Bruker D8 Advance X-Ray Diffractometer (XRD) using $\lambda$ = 1.5406 Å Cu K$\alpha$ radiation; the scanning interval was $10° \leq 2\theta \leq 90°$, and the scanning speed was 2°/min. The sample test results were analyzed with Jade 6 software.

## 3. Results and Discussion

### 3.1. Surface and Cross-Section Morphology

The electrodeposition technique utilized has a significant impact on the surface morphology and uniformity of the plated layer. The surface microscopic morphology of monopulse galvanized iron alloy and DC galvanized iron alloy plated at 3k and 30k magnification under SEM is presented in Figure 2. The monopulse zinc–iron alloy plating was produced utilizing process parameters of 30% duty cycle and 1000 Hz frequency, as depicted in Figure 2a–d. At the magnifications of 3k, 10k, 30k, and 50k, the plated layer exhibits a flat and smooth surface with a neat arrangement between the plated crystals and uniform particles, devoid of any defects, such as pinholes and impurities. One factor contributing to the acquisition of fine-grained deposits in pulsed electrodeposition is the occurrence of events during the toff period when the current is interrupted. This interval allows for the desorption of impurities and facilitates the renucleation process through the formation of smaller grains [40]. In contrast, the coating obtained through the DC plating process at the same magnifications, as depicted in Figure 2e–h, displays a rough and uneven surface characterized by a high porosity and a significant number of impurities.

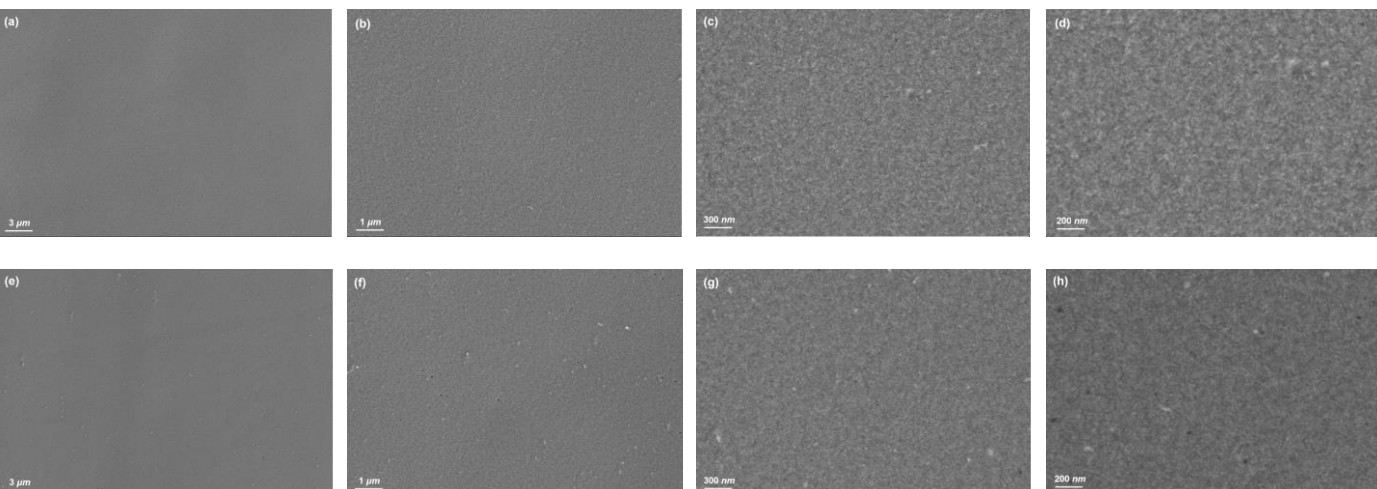

**Figure 2.** SEM-SE micrographs illustrating surface morphology of monopulse (**a**–**d**) and DC-plated (**e**–**h**) at 3k, 10k, 30k, and 50k magnifications, respectively.

The primary distinction between the morphologies achieved through monopulse and DC technology lies in the dimensions of the crystal clusters and grains. The coating produced by monopulse exhibits higher density, a more uniform structure, and agglomeration size, as well as a notably smoother appearance; this can be attributed to the elevated peak current in the monopulse process, which typically leads to an augmented nucleation rate and a diminished grain size in the coating. Additionally, the effects of $t_{on}$ and $t_{off}$ on the coating's characteristics are of great significance, as the adsorption–desorption phenomenon can exert a substantial influence on the crystallization process. The utilization of a 30% duty cycle in this investigation yields a desorption time that effectively eliminates impurities from the substrate surface and facilitates the nucleation crystallization of the coating. It is important to acknowledge that excessively low duty cycles, specifically prolonged $t_{off}$ times, can induce localized corrosion dissolution, thereby leading to defects in the coating structure [41,42].

Figure 3 illustrates that both the monopulse plating (Figure 3a–c) and the DC plating (Figure 3d–f) exhibit a monolayer coating structure from the cross-sectional perspective at the magnifications of 3k, 5k, and 10k. The monopulse plating demonstrates a more condensed crystal structure, decreased porosity, and finer grains when compared to the DC electroplating coating. As a result, the monopulse plating is more resistant to corrosion. Additionally, under identical average current density and working time conditions, the monopulse process yielded thicker coatings (10.14 μm and 7.24 μm) compared to the DC process. The interface between the coating and the substrate clearly indicates a stronger bonding force in the case of the monopulse process. At a high magnification of 10k, the monopulse process demonstrates a cohesive coating, whereas the cracks are observable on the coating obtained through the DC process at a magnification of 5k and become more pronounced at 10k magnification. The increased thickness of the coating and enhanced binding force will manifest at the macro scope as heightened hardness and superior resistance to wear. Consequently, this will effectively mitigate surface scratches, wear, or detachment, thereby substantially prolonging its lifespan.

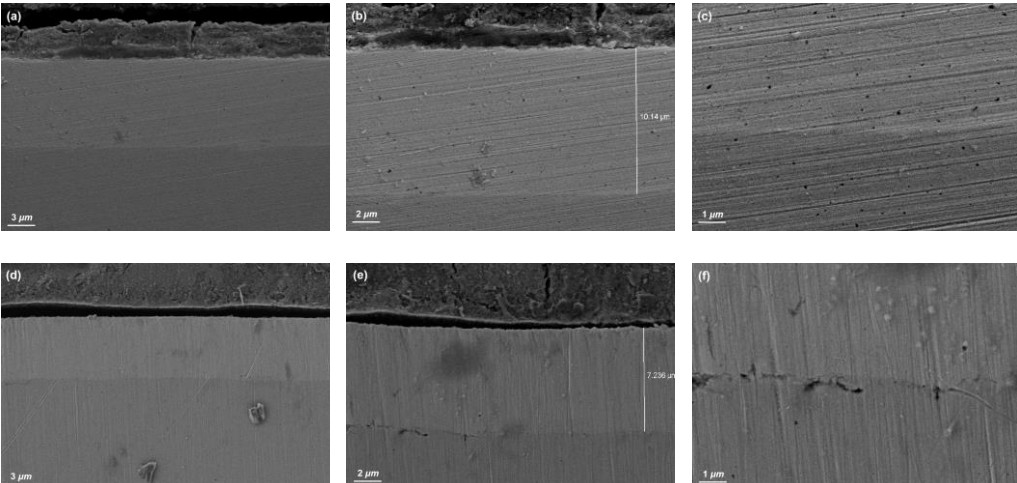

**Figure 3.** SEM-SE micrographs illustrating cross-section morphology of monopulse (**a**–**c**) and DC-plated (**d**–**f**) at 3k, 5k, and 10k magnifications, respectively.

The complexity of the phases in electrodeposited Zn–Fe alloys is attributed to the substantial influence of chemical composition [43,44]. At lower temperatures, the monoclinic Zeta phase ($FeZn_{13}$) constitutes approximately 6.5–5.2% of zinc–iron alloys [45,46] on the plating's phase structure. As a result, the reflectivity patterns of both the Zn-rich phase (ICDD: 00-004-0831) [47] and the $\zeta$ phase are observed in the corresponding deposits, as depicted in Figure 4. The accumulation of composites in the $\eta$(1 0 1) phase during monopulse plating is characterized by a notably weaker peak intensity, which may be attributed to the variation in the Zn–Fe content ratio across different plating methods. As the Fe content of the deposit varies, there is a gradual shift in certain peaks that correspond to specific crystal HKL planes towards either higher or lower 2θ angles.

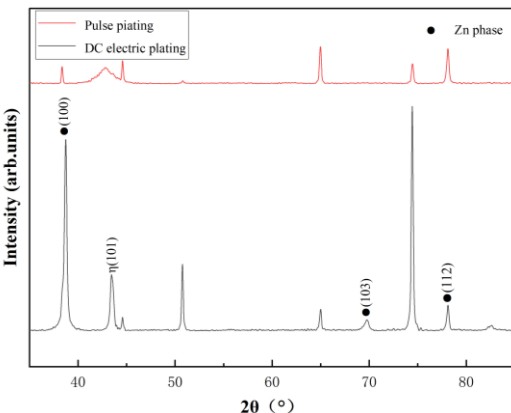

**Figure 4.** XRD diffraction pattern with the monopulse and DC plating method.

The crystal structure arrangement and morphology of Zn-based alloy deposits at a microscopic level can be influenced by the plating methods employed. These factors can subsequently impact the surface morphology of the deposit at a macroscopic level, ultimately affecting the mechanical properties and corrosion resistance of the deposit.

This study examines the notable disparity in crystal structure between coatings produced through monopulsed and DC processes, with a particular focus on the diffraction peak of the (1 0 1) phase. The full width at half maximum (FWHM) of this peak serves as a direct indicator of its shape, while the $W_{\mathrm{FWHM}}$ parameters establish a strong correlation with lattice defects. These defects encompass micro-deformation associated with disordered dislocations, as well as dislocations situated at grain boundaries and subgrain boundaries. The corrosion rate of the coating decreases as the $W_{\mathrm{FWHM}}$ value increases,

indicating that coatings with greater crystal structure defects exhibit enhanced corrosion resistance. This phenomenon can be attributed to the diminished crystal perfection, which heightens surface reactivity. The heightened Zn–Fe surface activity and altered metal structure may serve as precursors for the formation of an oxide film, which possesses superior protective properties [23,42,48].

### 3.2. Corrosion Resistance

An electrochemical workstation was utilized to conduct electrochemical testing on DC plating and monopulse galvanized ferroalloy coatings, obtained at a frequency of 30% duty cycle and 1000 Hz in a solution containing 3.5% sodium chloride. The Tafel curve of zinc ferroalloy plating with various plating methods is presented in Figure 5. The results in Figure 4 and Table 3 indicate that the corrosion potential of the DC galvanized ferroalloy coating is approximately −1031 mV. The pulse galvanized ferroalloy coating exhibits a corrosion potential of −1008 mV, while the monopulse plating demonstrates a higher corrosion potential than that of the DC galvanized ferroalloy. Table 3 presents the electrochemical corrosion current density of both monopulse and DC plating. Notably, the corrosion current density of the monopulse Zn–Fe alloy plating ($0.321 \times 10^{-5}$ A·cm$^{-2}$) is significantly lower than that of the DC plating Zn–Fe alloy ($3.122 \times 10^{-5}$ A·cm$^{-2}$). The corrosion current density ($i_{\text{corr}}$) serves as a direct indicator of the coating's corrosion resistance. Therefore, as the corrosion current density decreases, there is a corresponding enhancement in the corrosion resistance of the coating.

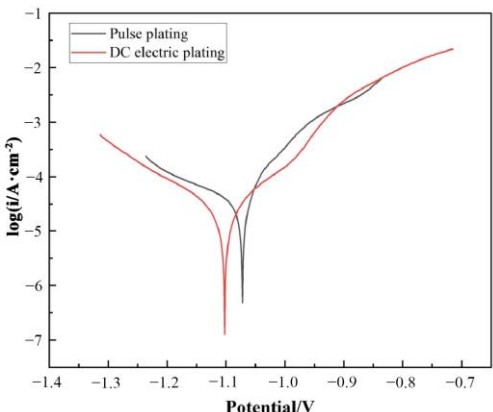

**Figure 5.** Tafel curve of zinc–iron alloy plating by monopulse and DC.

**Table 3.** Corrosion potential ($E_{\text{corr}}$), corrosion current density ($i_{\text{corr}}$), and polarization resistance ($R_{\text{p}}$) of samples with different surface treatment processes.

| The Surface Type | $E_{\text{corr}}$ (mV) | $i_{\text{corr}}$ (A·cm$^{-2}$) | $R_{\text{p}}(\Omega \cdot \text{cm}^2)$ | Corrosion Rate ($10^{-5}$ g·m$^{-2}$·h$^{-1}$) | Anodic Slope (mV/Decade) | Cathodic Slope (mV/Decade) |
|---|---|---|---|---|---|---|
| Monopulse plating | −1008 | $0.321 \times 10^{-5}$ | 898 | 0.386 | 7.899 | 8.183 |
| DC plating | −1031 | $3.122 \times 10^{-5}$ | 123 | 3.75 | 3.670 | 5.285 |

Furthermore, the corrosion rate, which is directly linked to the corrosion current density, can provide a more intuitive indication of corrosion resistance. The corrosion rate values for pulse and DC platings are $0.386 \times 10^{-5}$ g m$^{-2}$ h$^{-1}$ and $3.75 \times 10^{-5}$ g m$^{-2}$ h$^{-1}$, respectively. The disparity in corrosion rate between the two platings is of an order of magnitude, thus demonstrating that pulse plating exhibits superior corrosion resistance to the DC one. Compared with the traditional DC plating process, the use of monopulse plating Zn–Fe alloy technology significantly improves the corrosion resistance of the coating compared with DC plating.

The hypothesis that Zn polycrystalline grains with varying crystal orientations exhibit differential corrosion rates was proposed through an investigation into the corrosion of

zinc single crystals. The corrosion rate of zinc coating is influenced by its texture in a dual manner. The activation energy of the solution rises in tandem with the increase in packing density. Consequently, the surface with the highest density is expected to display the lowest corrosion rate. The packing density of the zinc crystal plane follows the order of $\rho (0\ 0\ l) > \rho (h\ k\ 0) > \rho (h\ 0\ 0)$. Furthermore, the development of a zinc oxide film on the surface of Zn is contingent upon the specific plane's index. Throughout the corrosion process, a pseudo-passive layer composed of zinc hydroxide and zinc oxide is generated on the coating's surface, effectively impeding any subsequent dissolution [11]. It has been observed that basal planes generate a thin oxide layer that exhibits a high level of protection, whereas other planes yield a thicker film with lower protective properties. Consequently, coatings featuring a low-index plane texture may exhibit enhanced stability owing to their increased metal atomic coordination and greater tolerance towards oxide films on these particular surfaces [49–51]. In this study, it is observed that the diffraction peak signals corresponding to the (1 0 1) and (1 0 3) planes of the monopulse coating exhibit significant attenuation when compared to those of the DC coatings. Based on the aforementioned analysis, it can be inferred that these two planes are prone to low-density accumulation, which consequently contributes to a higher corrosion rate and adversely affects the corrosion resistance of the coating. Consequently, the monopulse process employed in the production of the coating effectively mitigates the formation of the low-density accumulation phase, resulting in a reduced corrosion rate.

Figure 6 depicts the impedance curve following fitting, which was obtained through various plating methods. The presence of a semicircular feature in the impedance curve is indicative of a capacitance loop, and it is observed that the polarization resistance exhibits an upward trend as the diameter of the semicircle expands [52]. The semicircle is linked to the charge transfer process observed in ionic double-layer capacitors [53]. CPE encompasses various electrochemical phenomena that are contingent upon frequency, including double-layer capacitance and diffusion processes [54].

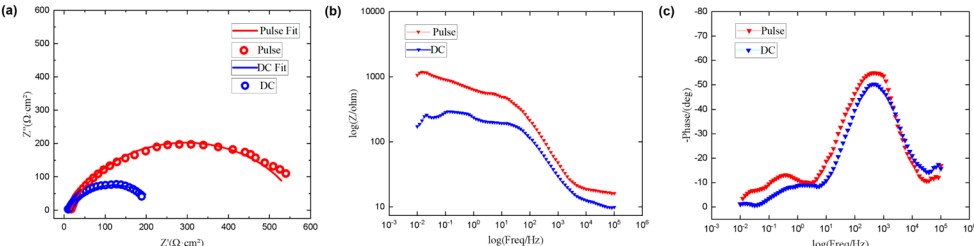

**Figure 6.** Electrochemical impedance spectroscopy (EIS) data, including Nyquist (**a**), Bode (**b**), and (**c**) diagrams.

The corresponding equivalent circuit diagram is presented in Figure 7, while Table 4 displays the fitting parameters. The semicircle response attributed to corrosion is evident in each curve. The uniformity of the solution system, testing equipment, and environment ensures that the starting point of both curves is identical. The relationship between the diameter of the semicircle and corrosion resistance is noteworthy. Specifically, in the context of monopulse plating, the Zn–Fe alloy plating displays a high arc radius and consequently exhibits high corrosion resistance. Table 4 presents the electrochemical impedance parameters obtained by fitting the impedance curve with the equivalent circuit. The equivalent circuit diagram utilized in the fitting is depicted in Figure 7, wherein $R_s$ represents the solution resistance, $R_{ct}$ signifies the charge transfer resistance, and CPE refers to the constant phase elements. Accordingly, a greater $R_p$ value signifies an elevated level of corrosion resistance in a coating [31]. Specifically, the $R_p$ value of the Zn–Fe alloy coating, which underwent monopulse plating, measured 561.50 $\Omega$, whereas the coating that underwent DC plating measured 223.10 $\Omega$. The substantial difference in $R_p$ values between the two coatings indicates that the monopulse electroplated Zn–Fe alloy coating exhibits superior corrosion resistance compared to the DC-plated coating.

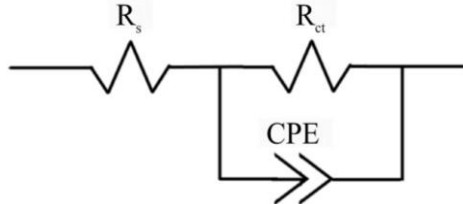

**Figure 7.** Equivalent circuit diagram of impedance diagrams fitting.

**Table 4.** Impedance spectra fitting and results of monopulse electrodeposition of Zinc ferroalloy and DC Zinc ferroalloy coatings (30% duty cycle, 1000 Hz frequency).

| The Surface Type | $R_s$ ($\Omega \cdot cm^2$) | CPE-T ($10^{-4}$ $\Omega^{-1} \cdot cm^{-2} \cdot s^n$) | $R_{ct}$ ($\Omega \cdot cm^2$) | $R_p$ ($\Omega \cdot cm^2$) |
|---|---|---|---|---|
| Monopulse plating [1] | 15.86 | 23.83 | 545.64 | 561.50 |
| DC plating [2] | 9.56 | 51.87 | 213.54 | 223.10 |

[1] The error% values of $R_s$ and $R_p$ from monopulse plating are 1.2395 and 1.7360, respectively. [2] The error% values of $R_s$ and $R_p$ from DC plating are 1.6895 and 6.1120, respectively.

*3.3. Composition Analysis*

The element surface distribution map (Figure 8) indicates a uniform distribution of all elements on the coating surface, with no evidence of segregation, except for defects. Figure 9 shows the microstructure and elemental distribution of the surface from DC-plated and monopulse-plated Zn–Fe alloy coatings. Table 5 displays specific elemental values for Figure 9, revealing a substantial quantity of zinc and trace amounts of Fe, O, C, and Na, among others. Notably, the elemental composition of the two distinct plating techniques differs significantly, with the monopulse plating method exhibiting a noteworthy reduction in iron content by 0.7%, resulting in enhanced corrosion resistance and leveling.

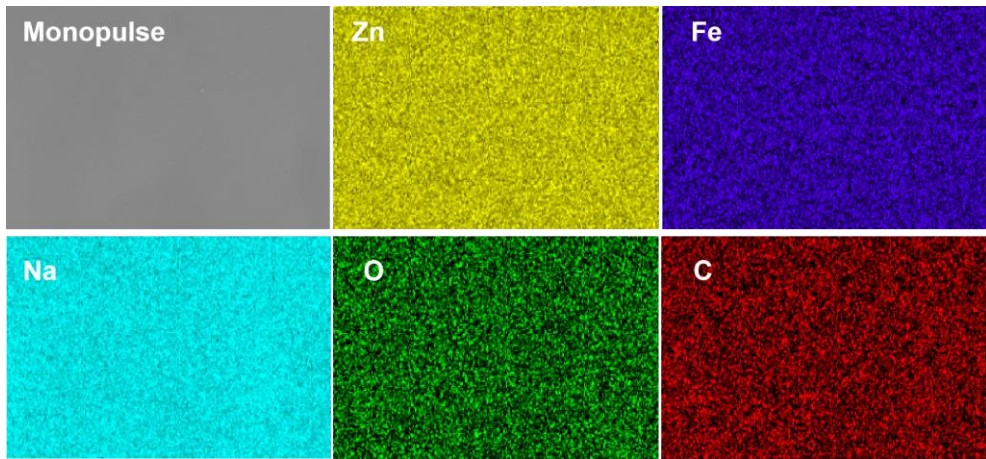

**Figure 8.** *Cont.*

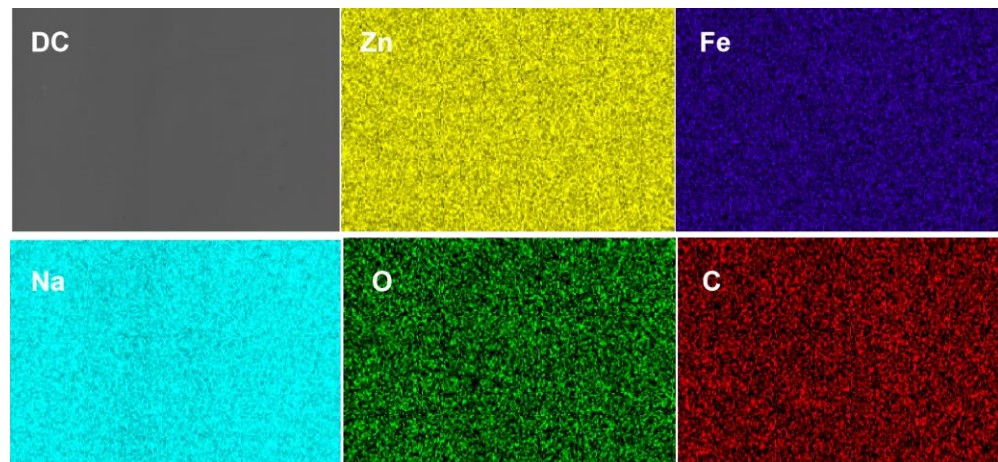

**Figure 8.** Energy dispersive spectroscopy (EDS) mapping for the surface of monopulse and DC coatings at 3k magnification.

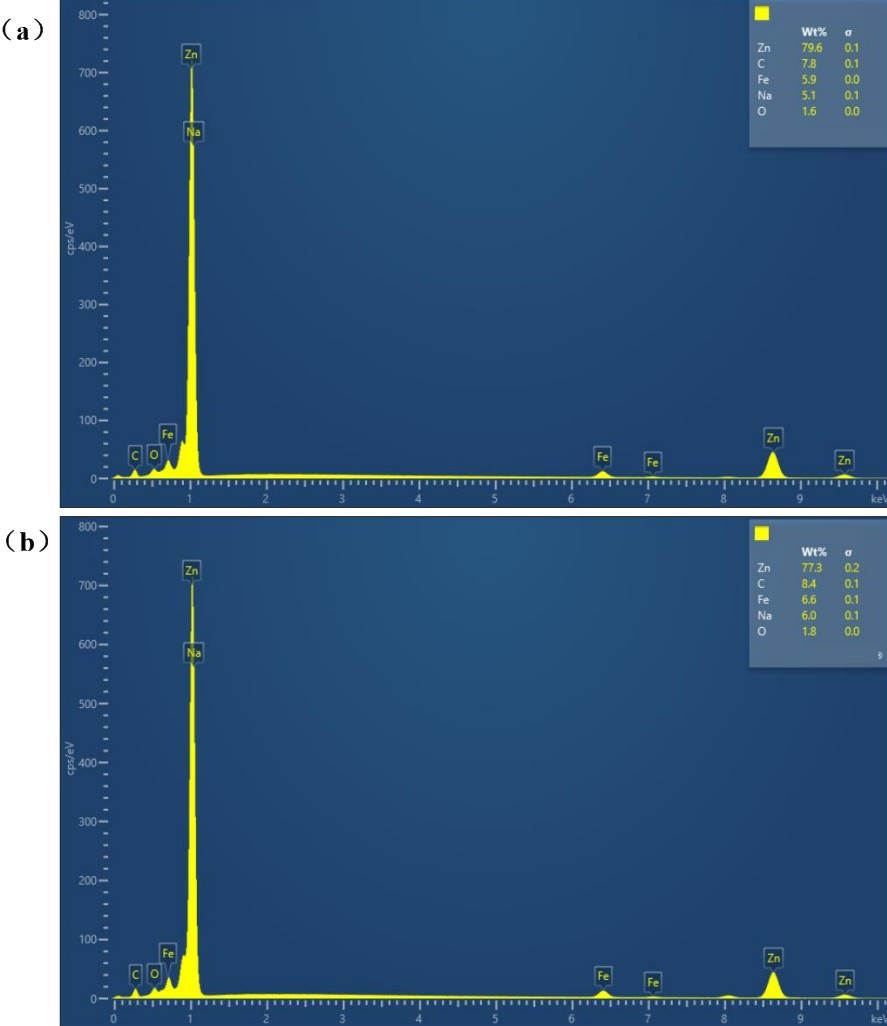

**Figure 9.** Energy dispersive spectroscopy (EDS) energy spectrum analysis: monopulse plating (**a**) and DC plating (**b**).

**Table 5.** Element content on the surface of monopulse and DC-plated coatings.

| Element | Monopulse (wt.%) | DC (wt.%) |
|---------|------------------|-----------|
| Zn | 79.6 | 77.3 |
| Fe | 5.9 | 6.6 |
| O | 1.6 | 1.8 |
| C | 7.8 | 8.4 |
| Na | 5.1 | 6.0 |

EDS was employed to examine the cross-sectional profiles of platings and their elemental distributions in two distinct electroplating techniques, as depicted in Figure 10. The elemental distribution clearly indicates the formation of a Zn-dominant coating between the mild steel substrate and the composite resin inlay. The diverse electrodeposition modes exert a substantial influence on the overall compactness and uniformity of the coating. The predominant constituents of the coating are Zn–Fe alloys, with the presence of the Na element being attributed to sodium ions derived from sodium hydroxide and ethylenediaminetetraacetic acid disodium salt present in the plating solution. The C and O elements are primarily sourced from the polymer in composite resin inlay, with the mild steel exhibiting negligible levels of the C element.

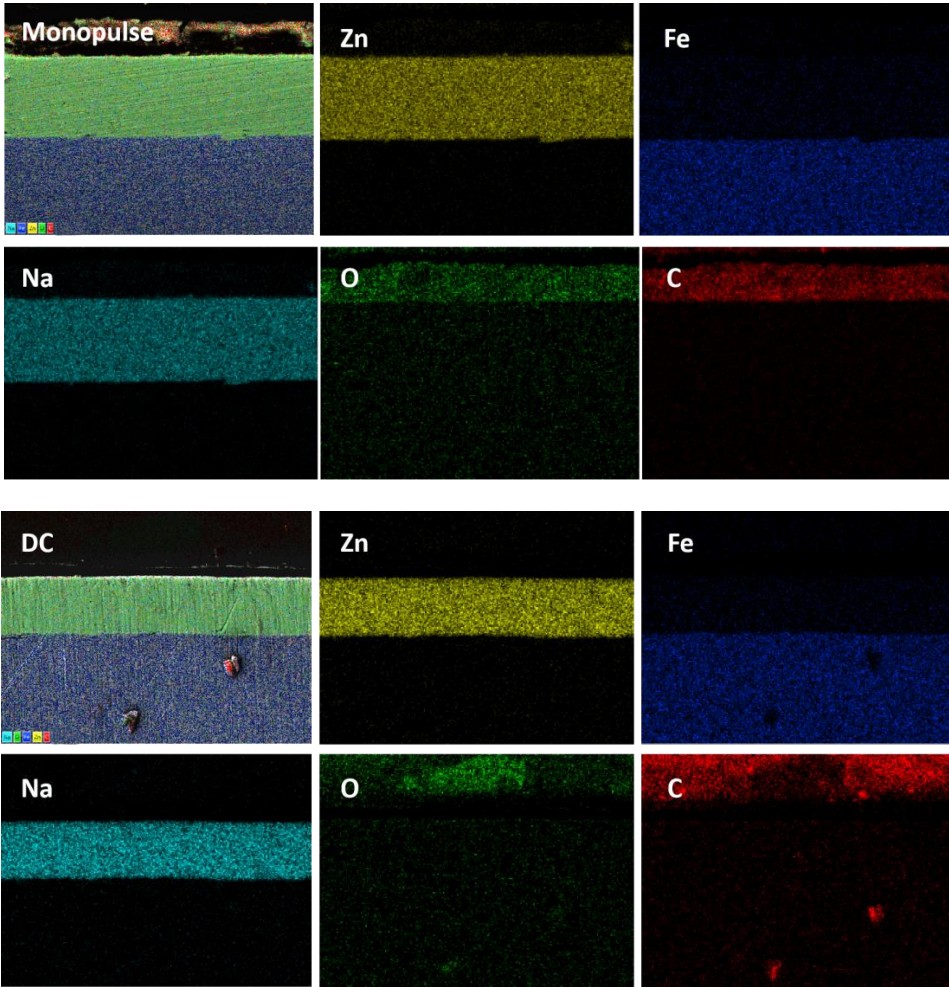

**Figure 10.** Energy dispersive spectroscopy (EDS) mapping for the cross-section of monopulse and DC coatings at 3k magnification.

There are discernible disparities in the cross-sectional morphology of the coating between the monopulse plating and the DC plating with identical bath composition. The

monopulse plating exhibits greater uniformity and smoothness, with an absence of defects, such as sagging and convexity, in contrast to the DC plating. This phenomenon can be attributed to the instantaneous high current density that occurs when the current is activated within a pulse period, leading to a reduction of metal ions under extremely high overpotential and a finer sedimentary layer grain. Upon cessation of the current, the discharge ions proximate to the cathode region revert to their original concentration, thereby obviating concentration polarization. This phenomenon facilitates the plating effect of the peak current in the subsequent period and also promotes the recrystallization, absorption, and desorption of the sedimentary layer during the shutdown interval. The maintenance of flat and uniformly thick plating is a crucial prerequisite for ensuring corrosion resistance. In comparison to DC plating, the pulse one exhibits superior physical and chemical properties, as well as electrochemical performance, as evidenced by the results of relevant tests.

## 4. Conclusions

In the context of the alkaline cyanide-free electroplating system, the plating process and plating solution formula were simultaneously modified, resulting in a notable enhancement of the coating's structure and morphology. Consequently, the physical and chemical properties of the coating were significantly improved.

Tafel results post monopulsed electrodeposition demonstrate a positive shift in the corrosion potential; the value increased from $-1031$ mV to $-1008$ mV, and the corrosion current density increased from $3.122 \times 10^{-5}$ A·cm$^{-2}$ to $0.321 \times 10^{-5}$ A·cm$^{-2}$, thus, substantiating this claim. The EIS impedance diagram indicates a substantial increase in the impedance radius of monopulse electroplating compared to DC electroplating, with an $R_\mathrm{p}$ value twice that of the latter, providing intuitive evidence for the superior corrosion resistance of the zinc–iron alloy utilized in monopulse electroplating.

CSurface and cross-section morphology and elements analysis evidenced that the Zn–Fe coating produced through pulsed current plating exhibits lower porosity, more compact crystal structure, and finer crystal particles when compared to the high porosity and defects observed on the surface of the DC coating. Additionally, the iron content in the monopulse coating was reduced from 6.6% to 5.9%. The application of monopulse electrodeposition has been shown to enhance the density and flatness of the resulting coating, surpassing that of DC electrodeposition in both macroscopic and microscopic characteristics.

The monopulse Zn–Fe coating has been observed to effectively diminish the presence of two low-density packing planes, namely (1 0 1) and (1 0 3), thereby mitigating the high corrosion rate associated with low packing density.

## 5. Patents

Both the patent [55] and this study are related to the same subject of our research team, with the latter featuring improved and optimized plating solution formula and process parameters.

**Author Contributions:** Conceptualization, F.C. and L.S.; methodology, F.C. and J.W.; software, J.W. and Y.W.; validation, F.C., J.W., X.W. (Xue Wang) and Y.W.; formal analysis, F.C., J.W., X.W. (Xue Wang) and A.S.; resources, L.S.; data curation, F.C., J.W., X.W. (Xue Wang), X.W. (Xiaomin Wang), Y.W. and Y.L.; writing—original draft preparation, F.C., J.W. and X.W. (Xue Wang); writing—review and editing, F.C., J.W., X.W. (Xue Wang) and L.S.; supervision, L.S.; project administration, F.C. and L.S.; funding acquisition, F.C. and L.S. All authors have read and agreed to the published version of the manuscript.

**Funding:** This work was financially supported by the Shandong Provincial Natural Science Foundation, China (Grant No. ZR202108100033), Shandong Province Science and Technology Small and Medium Enterprises Innovation Ability Improvement Project, China (No. 2022TSGC2561), and Youth Special Project of Qingdao Applied Basic Research Program (No. 18-2-2-70-jch).

**Institutional Review Board Statement:** Not applicable.

**Informed Consent Statement:** Not applicable.

**Data Availability Statement:** The data presented in this study are available in this article.

**Acknowledgments:** The authors want to thank the Jinan Institute of Quantum Technology for their assistance with the crystallographic elucidation and GIXRD measurements, respectively.

**Conflicts of Interest:** The authors declare no conflict of interest.

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
