# Peer review of "A Study on the Influence of the Electroplating Process on the Corrosion Resistance of Zinc-Based Alloy Coatings"

_coatings, doi:10.3390/coatings13101774_

Round 1

Reviewer 1 Report (New Reviewer)

The article discusses the use of monopulse electroplating as a technique for depositing zinc-iron alloy coatings and compares its performance to DC electroplating. Overall, the research highlights the advantages of monopulse electroplating in terms of grain size, uniformity, corrosion resistance, and potential environmental benefits.

Overall, the article shows promise in highlighting the advantages of monopulse electroplating, but some sections could benefit from additional detail and clarity. Addressing these points will improve the quality and readability of the manuscript.

1-It would be helpful to reference previous research or studies that have investigated similar topics. This can provide context for the significance of your findings and help readers understand how your work contributes to the field.

2-The article mentions various parameters involved in the monopulse electroplating process, such as peak current, duty cycle, and frequency. Could you provide more details on how these parameters were selected or optimized for the experiments?

3-It was mentioned that monopulse electroplating has the potential to reduce the usage of organic additives and mitigate environmental pollution. Could you elaborate on this aspect? How significant is the potential environmental benefit, and have you conducted any assessments in this regard.

4-If possible, consider enhancing the resolution of the figures or providing higher-resolution versions as supplementary material. Additionally, ensure that all figures are properly referenced and explained in the text to help readers interpret the data accurately.

The article discusses the use of monopulse electroplating as a technique for depositing zinc-iron alloy coatings and compares its performance to DC electroplating. Overall, the research highlights the advantages of monopulse electroplating in terms of grain size, uniformity, corrosion resistance, and potential environmental benefits.

Overall, the article shows promise in highlighting the advantages of monopulse electroplating, but some sections could benefit from additional detail and clarity. Addressing these points will improve the quality and readability of the manuscript.

1-It would be helpful to reference previous research or studies that have investigated similar topics. This can provide context for the significance of your findings and help readers understand how your work contributes to the field.

2-The article mentions various parameters involved in the monopulse electroplating process, such as peak current, duty cycle, and frequency. Could you provide more details on how these parameters were selected or optimized for the experiments?

3-It was mentioned that monopulse electroplating has the potential to reduce the usage of organic additives and mitigate environmental pollution. Could you elaborate on this aspect? How significant is the potential environmental benefit, and have you conducted any assessments in this regard.

4-If possible, consider enhancing the resolution of the figures or providing higher-resolution versions as supplementary material. Additionally, ensure that all figures are properly referenced and explained in the text to help readers interpret the data accurately.

Author Response

Reviewer 2 Report (New Reviewer)

Although the proposed manuscript is interesting, there are enough weaknesses that need to be improved. This based on the following:

·        The abstract should be reviewed again because it is very general and the objective is not clear.

·        The scope of the study is not well defined, the authors could better express it in the abstract

·        The introduction section is too long, there are 46 references out of 49 that the entire document has, this is incongruent

·        Line 163-174, the authors must improve the structure of the research objective

·        Section 2. materials and methods, must be restructured, very long paragraphs are used and confuse the reader

·        The authors should indicate how to obtain the electrodeposition parameters of the Zn-Fe alloy

·        Line 239: (section 2.3 Microstructural characterization), this subheading is wrong, it should be corrosion or electrochemical tests. Line 245: …test sample is 3.5 wt. % NaCl……

·        Line 240-241: The electrochemical techniques used are potentiodynamic polarization curves (using TAFEL extrapolation) and electrochemical impedance spectroscopy. The parameters used in each of the techniques must be indicated.

·        Line 245. From this line if it corresponds to the microstructural characterization; It must be indicated that the EDS composition is semiquantitative and is based on the SEM, it must be indicated which detector was used in the SEM for morphologies, secondary or backscattered electrons.

·        Figure 2:………….(d) SEM- SE o BES???

·        Table 3. The authors must indicate the values of the anodic and cathodic slopes. The values of the current must be in the same units in the graph and in the table of results.

·        The Nyquist diagram should be square on the axes, plot from 0 to 600 ohms on both axes to better understand the behavior. The Bode impedance diagrams must be integrated. The Nyquist diagram must have values of frequencies. The equivalent circuit does not correspond to the kinetics presented in the Nyquist diagram, it must be removed, there are not two elements of constant phase, in which case zoom in on the diagram.

·        Figure 7. Figure 7. Should be SEM-SE or BES surface micrograph....

·        The magnification of figures 7 and 10 must be indicated, this is independent of the micrometric scale. The images of figure 7 are blurred

·        The authors present an investigation where they only describe the results but the discussion supported by specialized literature is needed. It is important that the authors work hard in this section of discussion of results.

·        The conclusions are too extensive. Authors should review the conclusions and may be more specific.

·        The authors present 49 references.  

·         These authors contributed equally to this paper.????

Author Response

Reviewer 3 Report (Previous Reviewer 2)

The paper "Study on the influence of electroplating process on corrosion resistance of zinc-based alloy coatings" presents significant test results in terms of coatings on zinc alloys. The paper can be published in the Coatings journal after some minor updates:

1. In line 105, please add the influence of Zn in other types of materials, like biodegradable alloys, in terms of corrosion resistance. It also fits as a comparison. Suggested reference: 10.3390/ma16062487

2.line 250: add XRD parameters

3. Improve the contrast for figure 2.

4. For figure 3, add the ICDD file codes for phase compounds.

5. More explanation of the corrosion rate between mono plating and DC plating in terms of Zn and Fe influence

6. summarize the conclusions.

The rest is fine.

Author Response

Reviewer 4 Report (New Reviewer)

Dear Authors,

The fundamental goal of this research is to improve the formulation and methods of electrodeposited Zn-Fe alloy plating solution while also introducing a pulse procedure to fine-tune and optimize the coating characteristics. The primary salt used in electrodeposition is altered to reduce the influence of chloridion, and the primary and secondary brighteners are combined. Ascorbic acid is used to inhibit Fe2+ oxidation, while EDTA, which has excellent chelating characteristics, is used as the complexing agent to slow down the metal deposition rate. The coating has increased smoothness and a lower Fe concentration as a result of the use of pulse electroplating, resulting in improved corrosion resistance. To study the ensuing improvement in corrosion resistance, a comparative investigation of surface and cross-sectional morphology, alloy composition, coating thickness, and microstructural changes between monopulse and DC procedures was performed using several characterisation techniques.

I would still try to shorten the introduction, 2 and a half pages seem too much to me. The text is well-organized, with measurements and results reported clearly. To properly appreciate all of the details, the SEM pictures (figs 2 and 7) as well as figs 8 and 10 must be expanded larger.

Following these brief observations, I propose that the work be considered for publication in Coatings Journal.

Minor changes.

Round 2

Reviewer 2 Report (New Reviewer)

After reviewing the document, the authors did not address all the observations indicated in the first review, it is necessary to review the document again for the work to be accepted.

1. The Abstract must be improved, no results must contain values.

2. In the methods section, they must indicate the parameters used in electrochemical techniques such as potentiodynamic polarization; What was the sweep interval? Were the tests carried out in duplicate? Were ASTM standards or their equivalence used? ASTM G5 or ASTM G106??

3. In table 3 the values of the slopes must have units (mV/decade).

4. In figure 6, you must indicate which is the Nyquist diagram and which are the Bode diagrams, using sections (a) (b) and (c). The results changed completely, you should discuss your results against what already exists in the literature.

5. The authors should work more in the discussion of results, where there is no change in the document and there are no references (only 3 paragraphs, from reference 40 to 50)

6. In the conclusions there must be values of the research results

Author Response

This manuscript is a resubmission of an earlier submission. The following is a list of the peer review reports and author responses from that submission.

Round 1

Reviewer 1 Report

The manuscript under review possesses the potential to appeal to a specialized readership of this esteemed journal. Upon addressing the mentioned corrections in my review, it may be deemed suitable for publication.

  1. Figure 7 - Microstructures of the coating surfaces. It is recommended to revise the title of the figure to accurately reflect its nature as an elemental mapping rather than a microstructure. Please provide a comprehensive discussion regarding the distribution of elements over the surface within the manuscript.

  2. Furthermore, the authors have omitted cross-sectional investigations of the coatings in their submission. It is strongly advised to incorporate these studies into the manuscript. The inclusion of elemental mapping studies of the cross-sections would also prove beneficial.

  3. The primary concern lies in the absence of microstructural studies, including investigations of the phase composition. The manuscript's overall quality would be significantly enhanced by incorporating such studies as XRD, Raman spectroscopy, and TEM (STEM).

  4. It would be advantageous to include measurement errors in the figures, where feasible, for clarity and precision.

  5. The linguistic accuracy of the manuscript necessitates thorough scrutiny. Several instances of terminology misuse were observed. Consider substituting "obtaining" with more appropriate alternatives such as "deposition" or "production."

  6. The literature review pertaining to recently published articles on the same subject matter could benefit from improvement, as it suffers from a shortage of contemporary sources. The most recent citations date back to 2019. The authors may find the inclusion of the noteworthy articles [1-2] of value.

    References
    [1] Surf. Coat. Technol. V. 122, Is. 2-3, pp. 183-187.
    [2] Prot. of Metals and Physical Chemistry of Surfaces, V. 50, Is. 1, pp. 72-87.

  1. The linguistic accuracy of the manuscript necessitates thorough scrutiny. Several instances of terminology misuse were observed. Consider substituting "obtaining" with more appropriate alternatives such as "deposition" or "production."

Reviewer 2 Report

The paper "Study on the influence of electroplating process on corrosion  resistance of zinc-based alloy coatings " has some very interesting aspects regarding zinc-based alloy coatings, but some additional information needs to be added to the manuscript. The paper needs some minor corrections.

1. The introduction should be updated with some information regarding other types of coating techniques, like mimetic, APS, etc. Also, the influence of chemical elements should be added in terms of usage properties (microstructural, corrosion, mechanical characteristics)—literature review. Suggestions: 10.1016/j.apsusc.2015.05.111; 10.3390/mi12121447.

2. In 2.3. Microstructure Characterization Section, please add some specific characteristics of SEM analysis.

3. After SEM analysis, some XRD analysis should be added for phase characteristics.

4. Table 3 should be completed with corrosion rate values and comments in the text.

5. The reference list is short.

The rest is fine.

Reviewer 3 Report

Since there is no proper discussion on obtained results, this manuscript is not suggested for publication. Such a technical issue could be understood from the low number of references. Other comments could be seen as follows,

1) References must be updated and extended to 35 articles (at least), from published articles in 2017-2023. 

2) The novelty must be highlighted in the introduction, compared to the literature review.

3) All process parameters need references.

4) The etchant needs a reference.

5) The quality of SEM images is not proper. They must also be enlarged. 

6) No repeatability of testing can be seen in Figure 3. Moreover, the standard deviation must be added to the values in Table 3.

7) What was the testing standard?

8) The R2 value must be reported for curve-fitting in Figure 4. 

9) All features must be mentioned on SEM images in Figure 6, such as pits, etc. 

10) The scale bar must be provided for Figure 7.

11) The discussion is poor. Obtained results must be described and compared to other results of other articles. 

12) The conclusions part must be rewritten. One sentence is for the topic and highlighted results must be reported in bullets, one by one, to show the novelty. 

13)  What is the difference between this manuscript and the following articles?

*https://scholar.google.com/scholar?hl=en&as_sdt=2007&q=electroplating+process+on+corrosion+resistance+of+zinc-based+alloy+coatings&btnG=

Reviewer 4 Report

Abstract:

Electroplating solution formulation should be deleted from Abstract,  could be given in Experimental part, Materials and methods.

Other numbers given in Abstract also could be avoided, could be pointed out in Conclusion.

Please provide used mild steel characteristics.

Table 2. Parameters of the DC plating process., Please explain what is Title 2., Also add units for Temperature.

Figure 7. Microstructures of the coating surface and images of EDS element surface distribution. Please explain different colors, are they using different filters,  what purpose have the colored photographs, as they look the same, only color is different.

Table 5. Element content on the surface of Single-pulse plating coating and DC plating plating please, point out that the results presented in Table are from the Fig.7., Results of the EDS analysis.

Concclusion

The results show that compared with DC plating, the Zn-Fe alloy plating prepared 215 by pulse power single pulse plating has fewer defects on the surface of the plating, and 216 the plating layer at the defect has no cracks, and the surface is smooth and smooth com- 217 pared with the DC plating plating layer, and the grain is fine, which effectively improves 218 the leveling of the plating.

This analysis is not given in paper. Please provide microstructure analysis. No comments about defects is given in paper, but defects are mentioned in conclusion. Please delete this from conclusion, or provide analysis which could confirm the conclusion.

Abstract:

Electroplating solution formulation should be deleted from Abstract,  could be given in Experimental part, Materials and methods.

Other numbers given in Abstract also could be avoided, could be pointed out in Conclusion.

Please provide used mild steel characteristics.

Table 2. Parameters of the DC plating process., Please explain what is Title 2., Also add units for Temperature.

Figure 7. Microstructures of the coating surface and images of EDS element surface distribution. Please explain different colors, are they using different filters,  what purpose have the colored photographs, as they look the same, only color is different.

Table 5. Element content on the surface of Single-pulse plating coating and DC plating plating please, point out that the results presented in Table are from the Fig.7., Results of the EDS analysis.

Concclusion

The results show that compared with DC plating, the Zn-Fe alloy plating prepared 215 by pulse power single pulse plating has fewer defects on the surface of the plating, and 216 the plating layer at the defect has no cracks, and the surface is smooth and smooth com- 217 pared with the DC plating plating layer, and the grain is fine, which effectively improves 218 the leveling of the plating.

This analysis is not given in paper. Please provide microstructure analysis. No comments about defects is given in paper, but defects are mentioned in conclusion. Please delete this from conclusion, or provide analysis which could confirm the conclusion.

Round 2

Reviewer 2 Report

It is ok for publishing.

Author Response

We express our sincere gratitude to the reviewers for their invaluable guidance and assistance in the completion of this work. Appropriate grammatical changes and language corrections have been made.

Reviewer 3 Report

The following comments were not appropriately addressed. All answers to comments must be inert in the text. Only answering is not enough.

1) All process parameters need references. (What are the references?)

2) The etchant needs a reference. (No reference was added!)

3) The quality of SEM images is not proper. They must also be enlarged. (the quality has no changes!)

4) No repeatability of testing can be seen in Figure 3. Moreover, the standard deviation must be added to the values in Table 3. (The new figure must be added to the text! no standard deviation could be seen in Table 3!)

5)  What was the testing standard? (Not mentioned in the text!)

6) The R2 value must be reported for curve-fitting in Figure 4. (An Excel file could be used!)

7) All features must be mentioned on SEM images in Figure 6, such as pits, etc. (Not done!)

8) The discussion is poor. Obtained results must be described and compared to other results of other articles. (the results must be compared to other results of other articles!)

9) The conclusions part must be rewritten. One sentence is for the topic and highlighted results must be reported in bullets, one by one, to show the novelty.  (Not done! where are bullets?)

10) A new part for patents has no meaning. It must be mentioned in the research method. 

11) What is the difference between this manuscript and the following articles? (Not done!)

Author Response

We express our sincere gratitude to the reviewers for their invaluable guidance and assistance in the completion of this work. Appropriate grammatical changes and language corrections have been made.

In accordance with the recommendations of the editor, a number of significant inquiries have been revised and subsequently addressed. We extend our gratitude for the invaluable feedback offered by the reviewers.

Reviewer 4 Report

After  the revision process, and correction which authors provided, in present state paper is acceptible for publishing.

After  the revision process, and correction which authors provided, in present state paper is acceptible for publishing.

Author Response

(The authors gave the same response as above.)
